# Air Pollution in Urban Africa: Understanding Attitudes and Economic Valuation in the Case of Dakar, Senegal

**Sokhna Mbathio Diallo and Abdoulaye Seck \***

Economics Department, Cheikh Anta Diop University, Dakar 10700, Senegal
\* Correspondence: abdoulaye.seck@ucad.edu.sn; Tel.: +221-77-330-13-03

**Abstract:** The degradation of air quality is a real concern in Africa, as pollution levels are consistently above commonly accepted thresholds, and yet, very little is known about individual attitudes and the extent to which improved air quality is valued in the context of rapid urbanization. This research proposes to analyze the willingness to pay for reduced air quality in African urban areas. Using survey data from 427 individuals in Dakar (Senegal) between February and May 2019, a double-bound, dichotomous contingent valuation model is developed. The results first suggest that 70% of individuals are indeed willing to pay an average of CFA Francs 3114.8 (USD 5.6) per month to contribute to air quality improvement, and the corresponding value of a life year gained is estimated at CFA Francs 35,550.2 (USD 80) at least. The results also point to a great deal of heterogeneity in individual valuation schemes, as they vary with the perceptions of life expectancy gains, payment vehicles, and various individual characteristics, and also across months with different levels of temperatures. These findings could constitute important inputs into public strategies aimed at improving air quality in the African context.

**Keywords:** air pollution; contingent valuation; willingness to pay; value of one life year; Dakar

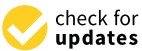



## 1. Introduction

Air pollution is increasingly among the greatest environmental threats to human health, and it reached the level of global health crisis. Indeed, it is estimated that 99% of the world population is living in places which inhabitability is significantly impaired by air pollution, as they did not meet the World Health Organization's (WHO) air quality guideline levels. For the most dangerous of the particles (2.5 microns or less in diameter, or $PM_{2.5}$), the guideline level was set to an annual mean of $5\mu g/m^3$ in 2021, below which the air is considered clean, with few impacts observed on human health. This new guideline level is half the previous level set in 2005, in response to "increases in quality and quantity of evidence of air pollution impacts" [1]. As a result of the many health-related issues that it is linked to, such as stroke, heart disease, lung disease, lower respiratory diseases (pneumonia), cancer, diabetes, and mental health diseases, air pollution is responsible for some 7 million premature deaths every year [1]. This health burden tends to disproportionately fall upon those living in developing nations, such as in Africa and West Asia.

In Africa, air pollution originating from fine particular matter ($PM_{2.5}$) reached a concentration of 43.29 $\mu g/m^3$, more than 8 times the WHO guideline level, and the second largest in the world, after South Asia (45.69 $\mu g/m^3$). The main source is windblown dust, especially for regions close to the deserts (such as Senegal). It accounts for the largest portion of $PM_{2.5}$ concentration, with 52.1% or a total of 22.57 $\mu g/m^3$. Next comes residential pollution, mostly from cooking and heating using biomass, and generating electricity from fossil fuels for homes, and transport, with 4.26 $\mu g/m^3$ or 9.8% [1]. The context of rapid urbanization, the quest for economic growth and development, as well as climate change or global warming that is shown to exacerbate the heath impact of pollution [2–4] suggest that air quality degradation and its associated consequences are meant to worsen, unless

economic and environmental processes are made compatible for efficient and sustainable development strategies.

Although these health-related consequences of air pollution and the generating mechanisms of air quality degradation tend to be very well known, most notably in developed countries where environmental awareness is relatively well advanced, relatively little is known in the socioeconomic context of developing countries where pollution levels are most critical, the health and economic impacts more severe, and the quest for industrialization and economic development more pronounced. If indeed the need to reconcile economic and environmental processes is a genuine concern of public policies, as suggested, for instance, by the commitment to the United Nations' Sustainable Development Goals (SDGs), the effectiveness of public policies and strategies remains uncertain, mostly because of the limited knowledge of popular support for climate interventions. This limited scope is most often due to the low involvement of the populations, for whom we ignore the mechanisms through which natural and environmental resources enter their preferences. Understanding attitudes and their valuation of environmental goods and services can thus be a means of assessing the scope of the public legitimacy of climate change policies as well as the extent of public involvement in their implementation.

With regard to air quality, policies aimed at its improvement may thus have legitimacy and public support, and the potential magnitude of these may be revealed by the value people place on the improved quality and the perceived benefits in terms of better health and living conditions. However, the limited research interest in people's attitudes towards air quality and in valuation patterns of its improvement in the African context, especially in urban settings, suggests that policy initiatives designed to combat air pollution and its effects are devoid of empirical evidence that would guide their design and implementation.

The objective of this research is then to analyze individual attitudes toward air pollution and their willingness to pay for reduced improved air pollution in African cities, with a focus on Dakar, Senegal, as a study context. More specifically, this research addresses the following questions: (i) To what extent does air quality fit into the preference and valuation patterns of Africa populations? (ii) How much are they willing to pay for improved air quality? (iii) What are the benefits in terms of increased life expectancy resulting from better living conditions? (iv) Beyond the average value, what are the mechanisms that generate possible heterogeneity in individual attitudes and valuation?

From a policy perspective, answering these questions could indicate the need for a public air pollution strategy in the dual context of global warming, rapid urbanization, and the quest for economic (growth) performance. It could also further legitimize public actions to combat air pollution if it turns out that a relatively large proportion of the population is willing to contribute significantly to policy financing and implementation, suggesting a credible and significant demand for air quality. The research output can contribute by telling about the efficient aspects of the policies in a context of constrained government financing and limited external contributions. The valuation exercise can indeed be an important part of a cost–benefit analysis of various policy interventions by informing about part of the associated benefits.

The overall objective of this research is to analyze individual attitudes towards air pollution and the extent to which improved air quality is valued in urban Africa, both in monetary terms and in terms of increased life expectancy in good health. This research contributes to the existing literature, the results of which seem to be most often dependent on the specific context of study, to the extent that the diversity, in terms of the incidence of air pollution and socio-economic environment, do not allow any generalization. Therefore, a context-specific study would be more useful, and it could feed into the emerging and ongoing stream of empirical knowledge.

The methodological approach is based on the Contingent Valuation Method (CVM) in its double-bound, dichotomous format, and the data are derived from an in situ survey of a sample of 427 individuals representing the Dakar region across its four administrative departments. The survey consisted of four waves, corresponding to the 4 months of

February–May 2019, as a way of capturing any potential seasonal variability of air pollution. In effect, over that period, temperature rises and the country experiences frequent episodes of dust, both having consequences on the level of pollution and its associated hazardous impacts, and by ricochet, individual attitudes towards air quality.

The rest of the paper unfolds as follows: Section 2 presents the context of the study; Section 3 describes the methodological approach and the survey data; Section 4 presents and discusses the findings; and Section 5 concludes.

## 2. Study Context

Senegal has a Sudano-Sahelian climate, sub-tropical in the south and semi-desertic in the center and the north. It is characterized by the alternation of a dry and mild season from November to mid-June and a wet and relatively hot season from mid-June to October. During the dry season, winds from the Sahara Desert, known as the harmattan, invade the country and cover it with a relatively thick coat of dust.

The climate in metropolitan Dakar is warm to hot all year round, with an annual average of 20.9 degrees (source: https://www.worlddata.info/africa/senegal/climate-dakar.php (accessed on 12 October 2022)). This is the lowest in the country, mostly due to its coastal position. The region is only 0.28% of Senegal's landmass, but with is home to nearly a quarter of Senegalese population of more than 16 million [5]. In addition to being the political capital of the country, it is also the administrative and economic center, making it the most advanced region in the country.

Moreover, the establishment of large industrial plants, near the autonomous port, all along the bay of Hann to Rufisque, makes Dakar the largest industrial and product processing center in the country. It accounts for 80% of industrial activities and is one of the main destinations for internal and external migrants. In addition, more than 70% of freight trips originate or depart from Dakar [6] (ANSD: *Agence Nationale de la Statistique et de la Démographie* (National Agency for Statistics and Demography)). This is facilitated by the economic weight of the Dakar as well as improved highway infrastructure.

This economic and industrial stance of Dakar resulted in significant sanitation and environmental issues, as pressure to existing infrastructure tends to grow with somehow unfettered urbanization. For instance, when it comes to solid waste, as of 2021, it is estimated that the region produces some 2684.5 tons per day (or 979,842 tons per year). This represents more than a quarter of the national daily volume of around 8664.4 tons (source: https://www.giz.de/de/downloads/SectorBrief_Senegal_Waste.pdf (accessed on 14 October 2022)).

As far as air quality is concerned, pollution levels tend to go beyond the thresholds commonly accepted by the WHO, and even those set at a national level. For the latter, the Air Quality Index (AQI) covering five main pollutants regulated by the law on air quality (surface ozone, dust particles, carbon monoxide, sulphur dioxide, and nitrogen dioxide) is developed by the Center for Air Quality Management (CGQA), which is under the supervision of the Ministry of the Environment and Sustainable Development (MEDD). Its mission is to monitor ambient air pollution, to inform the public on the state of air quality, to provide the State with reports on air quality for decision making, and to contribute to the promotion of air quality, among others. The Center has a network of six fixed monitoring stations, as well as a mobile station (truck) acquired by the DEEC in 2005.

The AQI averages various sub-indices corresponding to each of the main pollutants. On a scale of 0–300, with higher values corresponding to higher pollution level, four subdivisions are considered, each one represented by a color code: "green" meaning "good" air quality, with scores under 50; "yellow" indicating "acceptable" air quality, with however some possible risks for very sensitive persons to pollutants, such as ozone; "orange" being synonymous with "bad" air quality, typically for scores ranging from 100 to 200, and people with respiratory illnesses and other health-related issues being negatively impacted; and "red" for "very bad" air quality, which should prompt health emergency as a larger share of the population is seriously impacted. More details can be

found here: https://www.denv.gouv.sn/indice-de-la-qualite-de-lair-iqa/ (accessed on 14 October 2022).

Information is summarized through the IQA and is then disseminated every day by the CGQA to the general populations through various means: an electronic messaging system to a group including the Ministry of Health and interested organizations and individuals, some media outlets, as well as the MEDD website and social networks. This is part of an overall strategy to inform and sensitize the populations and raise awareness about the health issues associated with air pollution.

Figure 1 shows the annual trends in air quality as measured by the AQI over the recent years. For a typical year, a cyclical or seasonal pattern of air quality emerges. The beginning of the year corresponds to pollution peaks. This is November–June, during which the climate is marked by the dry harmattan wind with repeated episodes of sand dust that cover a significant part of the country, including Dakar. From the middle of the year until October, the country experiences rains that dissipate most of the pollutants, especially suspended particular matters.

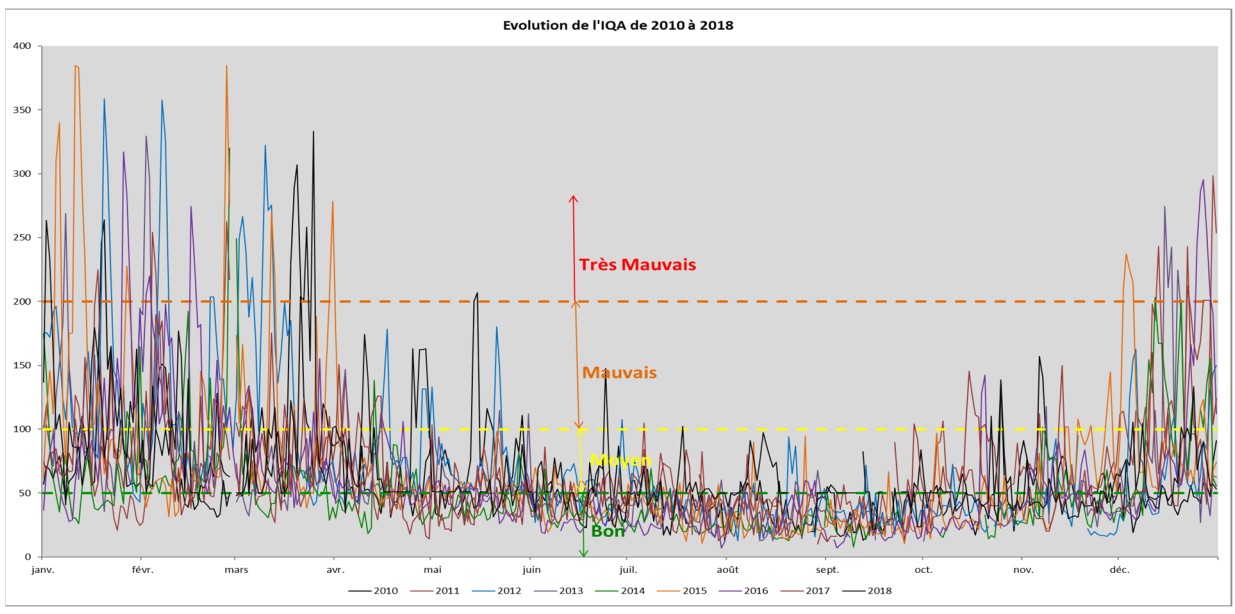

**Figure 1.** Evolution of Air Quality Index (AQI) in Dakar. Source: 2018 Annual Report, Dakar Air Quality Management Center. Notes: "Très Mauvais" translates to "Very Bad" level of pollution; "Mauvais" to "Bad"; "Moyen" to "Average"; and "Bon" to "Good" air quality.

Globally, it is estimated that 93% of children live in environments with air pollution levels above the thresholds set by the WHO. Moreover, the majority of deaths in children are due to particular matter pollution occurring in low- and middle-income countries. Indeed, the mortality rate attributable to air pollution in low- and middle-income African countries per 100,000 children worldwide is 184 [7]. In Senegal, between 2 and 5% of children are born prematurely due to the concentration of fine particles in the air and many of them are born with asthma. Asthma can be greatly aggravated by pollutant emissions [7]. In 2014, the Ministry of Health and Social Action estimated the number of patients with asthma to be 7000 in Senegal, which is more than double the figure in 2011. More recently (2018), it sought to affect 3% of children and 6% of adults, and environmental factors are considered to be the main factors contributing to this relatively high incidence (source: https://www.seneplus.com/sante/le-taux-de-prevalence-de-lasthme-estime-10-au-senegal (accessed on 14 October 2022)).

This pollution is estimated to cause more than 50% of acute respiratory infections in children under the age of five in low- and middle-income countries [7]. In Africa, two-thirds of children (66%) are concerned. In addition, the survey conducted by the national

statistical agency (ANSD) on the economic and social situation in 2018 showed that 59% of children with acute respiratory infections were taken to health facilities. The same year, the prevalence rate of cardiovascular diseases was 17% and for chronic respiratory conditions it was 3% [7]. A study looking into the externalities associated with transports in Dakar estimated that a total of 2.1 million individuals were impacted by air pollution, and the cost amounted to CFA Francs 63 billion (or USD 114.5 million), representing 2.7% of the country's GDP [8]. However, given the focus on transport as the source of emissions (for pollutants such as $PM_{10}$ and lead), one could assume that the actual total cost of air pollution that would account for all emission sources would be much higher. In the absence of the most recent study, it is not known the extent of the economic cost of air pollution, as favorable policy initiatives (age limitation of imported vehicles and prohibition of lead in gasoline) occurred in the context of increased demand for vehicles.

Air pollution in Dakar has different sources of emissions, according to ANSD (2020). These are mainly industrial, transport system, and greenhouse gas emissions.

– Industrial sector: During the year 2018, this sector shows an upward trend that began in 2016. This increase is due to the good performance of the chemical, extractive, energy, and manufacturing industries, which are major sources of air pollution. Indeed, 90% of electricity generation (638,000 kW in 2010) comes from fossil fuels and the remaining 10% comes from renewable energy sources [9].

– Transports: Road transport accounts for 95% of passenger transport in Senegal, with a fleet of 820,289 vehicles in 2018 compared to 766,737 in 2017, a 7-percent increase. It is heavily concentrated in the Dakar region, which is home to 74% of the national fleet, the bulk of which being second-hand vehicles. In addition, the fleet is relatively old, with an average of 15 years of existence. It is predominantly diesel-powered vehicles [9], which contribute to pollutant releases that exceed WHO standards and further affect the health of individuals.

– Greenhouse gas (GHG) emissions: The agriculture sector is the largest contributor of GHG with 49% of total emissions, followed by the energy sector (40%) and waste and industry sectors (7% and 4%, respectively). Waste treatment by either burning or storage results in the release of GHG into the air. In addition, the percentage of Senegalese using solid fuel for cooking, such as charcoal or wood, represents 74.4% nationally, and only 24% using relatively cleaner energy (electricity and gas for instance). However, the implementation of the Clean Development Mechanism (CDM) project sought to prevent the release of 298,424 tons of carbon dioxide, compared to 645,544 tons in 2017.

To further combat air pollution and its adverse impacts on health and the economy, the government implemented regulation on industrial emissions in the Senegalese environmental code and promoted renewable energies in the 2012–2017 Energy Sector Policy Letter, mostly in accordance with the SDGs. Specific policy initiatives include a limitation on motor vehicle exhausts, elimination of lead in gasoline since 2005, and age limitation for imported vehicles (first set at 5 years in 2008, and then raised to 8 in 2012).

Despite these various programs and laws put in place, the environment sector is still faced with several issues, most of them having to do with resource management and air pollution control, to which solutions must be found in order to achieve the SDGs. In addition, a relatively limited understanding of the demand side of environmental preservation and quality, in terms of individual attitudes, preferences, and valuations, tends to limit the scope, effectiveness, or even the legitimacy, of the various measures aimed at combating air pollution.

## 3. Methodology and Survey Data

This section presents the empirical model, discusses the data source and provides some descriptive statistics in the analysis of individual attitudes towards air pollution and the economic valuation of improved air quality.

### 3.1. Modeling Air Quality as a Contingent Good

Conceptually, analysis of the valuation of natural asset improvement starts with a random utility function [10–13]. The utility of a given individual faced with a choice over different alternatives has two components: a deterministic part that reflects the characteristics of the alternative and of the individual, and a random component, which includes unobserved factors, among others.

For a given person $j$, the indirect utility function is defined as:

$$U_{ij} = (Y_j, m_j, \varepsilon_{ij}) \tag{1}$$

where $i = 1$ corresponds to the situation after implementation of the program and $i = 0$ to the status quo, $j$'s income is $Y$, $m$ is the vector of $j$'s characteristics (age, gender, education level, etc.), and $\varepsilon_{ij}$ the error term. $Y_j$ and $m_j$ are the subsets of the characteristic vector $X$ of $j$.

As a result of air quality improvement, the individual will then move from the initial state of status quo $Z_0$ to the hypothetic scenario of the pollution-free state $Z_1$, with $Z_1 > Z_0$ in terms of air quality. In order to achieve the $Z_1$, the individual is required to make a financial contribution that would reflect its perceived gain. In essence, he/she is asked to pay for improved air quality. He/she will do so only if the shift corresponds to a utility gain, that is:

$$U_1(Y_j - P_{1j}, m_j, Z_1, \varepsilon_{1j}) >, m_j, Z_0, \varepsilon_{0j} \tag{2}$$

The probability of answering in the affirmative is then:

$$\Pr(yes_j) = \Pr\left[U_1(Y_j - P_{1j}, m_j, Z_1, \varepsilon_{1j}) > U_0(Y_j, m_j, Z_0, \varepsilon_{0j})\right] \tag{3}$$

Following [13], it is assumed that the random utility function is both additive and separable in its deterministic and stochastic components; that is:

$$U_i(Y_j, m_j, Z_i, \varepsilon_{ij}) = V_i(Y_j, m_j, Z_i) + \varepsilon_{ij}$$

The probability that the individual accepts to pay for improved air quality, or answering "yes" to the price offer) is then given by:

$$\Pr(Yes_j/Y_j, m_j) = \Pr\left[V_1(Y_j - P_{1j}, m_j, Z_1) + \varepsilon_{1j} > V_0(Y_j, m_j, Z_0) + \varepsilon_{0j}\right] \tag{4}$$

By considering a linear form of the deterministic component, that is, $V_{ij}(Y_j, m_j) = \alpha_i m_j + \beta_i Y_j$, it then comes:

$$V_{1j} - V_{0j} = m_j(\alpha_1 - \alpha_0) + (Y_j - P_{1j})\beta_1 - Y_j\beta_0 \tag{5}$$

Equation (5) is the classical expression of the surplus. It is the measure of the change in the individual's welfare due to an improvement in air quality. From one individual to another, heterogeneity in the levels of valuation could arise from differences in income or prices, or from individual characteristics. Empirical analysis will then be able to indicate the levels of individual valuation of air quality, the extent of any heterogeneity, as well as the generating factors.

From an empirical perspective, a double-bound contingent valuation method is considered. It consists of offering a first price $P_{1j}$ to the respondents, and then, depending on the affirmative or negative response, introducing a second price $P_{2j}^o$ that is higher or lower than the first. Compared to the single bound approach, which offers only one price, introducing a second price is shown to add more precision in the valuation exercise [14]. In addition, to correct for the anchoring bias, the first offer will be varied across respondents.

Given the number of two dichotomous responses (yes or no) and the possibility that the first response can influence the second one, a bivariate probit model is then considered. It allows to simultaneously model dichotomous qualitative events while accounting for any dependency. Separate models will suffer for biased estimates by ignoring such dependency.

If both answers to the bids are affirmative ("yes" to both), then the individual's WTP is greater than $P_{2j}^o$, and it comes to the corresponding probability:

$$\Pr\left(yes_j, yes_j\right) = \Phi\left(\frac{\alpha}{\sigma}m_j - \frac{\beta}{\sigma}P_{2j}^o\right) \tag{6}$$

One can similarly derive the probability of the remaining 3 possible combinations of responses:

$$\Pr\left(yes_j, no_j\right) = \Phi\left(\frac{\alpha}{\sigma}m_j - \frac{\beta}{\sigma}P_{1j}\right) - \Phi\left(\frac{\alpha}{\sigma}m_j - \frac{\beta}{\sigma}P_{2j}^o\right) \tag{7}$$

$$\Pr\left(no_j, yes_j\right) = \Phi\left(\frac{\alpha}{\sigma}m_j - \frac{\beta}{\sigma}P_{2j}^n\right) - \Phi\left(\frac{\alpha}{\sigma}m_j - \frac{\beta}{\sigma}P_{1j}\right) \tag{8}$$

$$\Pr\left(no_j, no_j\right) = 1 - \Phi\left(\frac{\alpha}{\sigma}m_j - \frac{\beta}{\sigma}P_{2j}^o\right) \tag{9}$$

The likelihood function is then:

$$
\begin{aligned}
&L(\alpha, \beta, \sigma, Y, m, P) \\
&= \sum_{j=1}^{N}\left\{d_j^{1,1}ln\left[\Phi\left(\frac{\alpha}{\sigma}m_j - \frac{\beta}{\sigma}P_{2j}^o\right)\right] + d_j^{0,0}ln\left[1 - \Phi\left(1 - \Phi\left(\frac{\alpha}{\sigma}m_j - \frac{\beta}{\sigma}P_{2j}^o\right)\right)\right] \right. \\
&\quad + d_j^{0,1}ln\left[\Phi\left(\frac{\alpha}{\sigma}m_j - \frac{\beta}{\sigma}P_{2j}^n\right) - \Phi\left(\frac{\alpha}{\sigma}m_j - \frac{\beta}{\sigma}P_{1j}\right)\right] \\
&\quad \left. + d_j^{1,0}ln\left[\Phi\left(\frac{\alpha}{\sigma}m_j - \frac{\beta}{\sigma}P_{1j}\right) - \Phi\left(\frac{\alpha}{\sigma}m_j - \frac{\beta}{\sigma}P_{2j}^o\right)\right]\right\}
\end{aligned} \tag{10}
$$

With $d_j$, there is a set of indicator variables whose values relate to actual responses to offers by individuals $j$ (1 = yes and 0 = no).

The maximization of this function provides an estimate of the parameters and an average and median willingness to pay. This method increases the statistical efficiency. It is shown that the introduction of a second price increases the efficiency of the estimation of the model parameters and the willingness to pay [14].

The vector of individual and household characteristics, $m_j$ includes age, gender, education, income, household head status, perceived life expectancy gain (3, 6, and 9 months) as a result of air quality improvement, payment mode (electricity bill tax, consumption tax, fuel tax, and voluntary contribution), pollution-related illness for the individual or a family member, perception of air quality, household size, proportion of children under 5, month or seasonality (corresponding to the survey period—February to May), etc.

Accounting for the non-responses (or those who did not accept to pay) is an important part of the modeling strategy. The first approach will consist of ignoring them and run a bivariate probit model (assuming normal distribution of the errors), with the risk of running into a selection bias. If the later turns out to be the case, then the estimation will not only be biased and inconsistent, but they will more likely overestimate the WTPs (to the extent that non-responses correspond to genuine zeroes). The alternative approach consists of actually accounting for the non-responses, and modeling the probability of positive values within a Heckman-type framework. The inverse Mill's ratio can then be obtained and inserted as an explanatory variable in the regression of the bivariate probit model. If this variable turns out insignificant, this will then be an indication of no selection bias, and we would consider the initial modelling approach (i.e., without selection).

The average WTP is the measure that allows us to calculate the value of one life year (VOLY), following [15]. VOLY is calculated as the present value, with the assumption of a zero discount rate, over their remaining lifetime. We will have the average WTP reported per month per individual multiplied by 12 to get the annual WTP. The average WTP values are proportional to the life expectancy gained. That is, the ratio of 12 months/6 months is 1.9, approximately equal to 2. This can be explained by the fact that individuals see the difference between the risk of increasing life expectancy between 6 months and 1 year as an extension of life [15]. Since we have the annual WTP for a gain in life expectancy of 3, 6,

and 9 months, this number is multiplied by the annual conversion factors 4, 2, and 12/9, respectively, to obtain the WTP for a year of life gained.

$$VOLY = (WTP_3 * 12) * 4 * \Delta EV \tag{11}$$

$$VOLY = (WTP_6 * 12) * 2 * \Delta EV \tag{12}$$

$$VOLY = (WTP_9 * 12) * \left(\frac{12}{9}\right) * \Delta EV \tag{13}$$

*3.2. Survey, Data and Descriptive Analysis*

The study context of the Dakar region is made up of four administrative units or departments: Metropolitan Dakar, Rufisque, Guediawaye, and Pikine. The population sampling technique is twofold. First, the representativeness at the department level requires that the relative size of each subsample follows its share in the total population of the region. The total population of 3.52 million in the region is distributed as follows: Metropolitan Dakar (36.5%), Rufisque (15.6%), Guediawaye (10.5%), and Pikine (37.3%). Second, individuals are selected at random across each department and on weekends to ensure that the diversity of the population is captured (i.e., between the employed and the non-occupied). The sample size takes into account the population under consideration, the desired level of precision (5% error), the non-response rate (assumed to be 10%) and the effect of the sample design (1).

The questionnaire is structured in a standard way and consists of the following three parts: (i) a description of the (contingent) service to be evaluated, ranging from the general air quality status in Dakar to personal perceptions of air quality and the scenario of expected life expectancy gain when air quality is improved; (ii) the evaluation itself, which is based on the double-bound, close-ended question format; and (iii) the socio-economic characteristics, such as age, gender, education, income, etc.

To avoid the inking bias associated with the absence of a reference for the choice of prices, three initial offers are considered: CFA Francs 2000, 3500, and 5000. These prizes are offered randomly to individuals. The double-bound format means that a second price offer is proposed, higher than the first if the response to the first is affirmative and lower in the opposite case. These price offers are subject on the one hand to a gain in life expectancy associated with better living conditions due to improved air quality, and on the other hand to various modes of payment, ranging from taxes to voluntary donations. For life expectancy, three potential gains were proposed (randomly): 3, 6, and 9 months, and the investigator also proposed one randomly.

A pretest was conducted one week prior to the survey with a group of students. It focused on the script and content of the willingness-to-pay questions. The actual survey was conducted monthly between February and May 2019. In total, a sample of 427 people was obtained. It is representative of the population of Dakar through its four departments.

Detailed descriptive statistics are provided in Appendix A (for the whole sample and all variables), and Appendix B (distribution of positive willingness to pay). Some 70% responded positively to the question of "Are you willing to contribute financially to any policy that seeks to improve air quality in Dakar?" Among the remaining 30% who are unwilling to contribute, 29.3% indicated being against the very principle of having to pay. This could be the typical case of protest values, and not real zeros, as individuals may consider that they should not be the ones to bear the cost of pollution they did not contribute to. For the rest (70.7% of those unwilling to contribute, or 21.2%, or one-fifth of the total population), one could consider that these are the cases of genuine zeros, representing individuals who do not attach any value to air quality. Overall, this gives a first indication of the sensitivity of the large majority of the Dakar residents (close to 80%) to the quality of the air they breathe and their desire to see it improved.

Further analysis of the distribution of positive valuation shows those individuals tend to be younger, more educated, with a higher perception of air quality. They also tend to

be occupied and enjoy higher income, although they are also more likely to suffer from pollution-related illness. Gender appears not to be a significant characteristic or discriminant. Additional details regarding the distribution of individuals willing to contribute are provided in Table A2 in the Appendix B. As indicated before, the Probit regression analysis will go further to reveal the main characteristics of these individuals as opposed to factors affecting the likelihood that any given individual is willing to pay or not.

Table 1 below provides a description of the responses to the price offers. For the first offers, the proportion of positive responses decreases with the level: nearly two-thirds for the price of CFA Francs 2000, and a little more and a little less than 1/3 for the prices of CFA Francs 3500 and CFA Francs 5000. We find that when the price increases, demand tends to decrease. This is consistent with the literature, as the law of demand states that an increase in price makes the good less attractive to consumers.

**Table 1.** Responses to the bid sequences (in CFA Francs).

| First Offer | Yes (Count and %) | | No (Count and %) | | Second Offer | Yes (Count and %) | | No (Count and %) | |
|---|---|---|---|---|---|---|---|---|---|
| 2000 | 85 | 62.0% | 52 | 38.0% | 1000 | 36 | 69.2% | 16 | 30.8% |
| | | | | | 3000 | 45 | 52.9% | 40 | 47.1% |
| 3500 | 33 | 37.5% | 55 | 62.5% | 2500 | 35 | 62.5% | 21 | 37.5% |
| | | | | | 4500 | 14 | 43.8% | 18 | 56.3% |
| 5000 | 25 | 32.9% | 51 | 67.1% | 4000 | 16 | 32.0% | 34 | 68.0% |
| | | | | | 6000 | 22 | 84.6% | 4 | 15.4% |
| Total | 143 | 47.5% | 158 | 52.5% | Total | 168 | 55.8% | 133 | 44.2% |

Source: Authors' calculations based on data from a survey they conducted in 2019.

However, when considering the second price, the responses given by the subgroup of respondents who accepted the first price do not seem to show an upward or downward trend. On the other hand, for those who answered "no" to the first offer, the responses to the second offer seem to be close to the law of demand: the proportion of acceptance decreases with the price level.

Moreover, while on average respondents are more likely to refuse the first price offers than to accept them (47.5% vs. 52.5%, all prices considered), they tend to be more favorable to the second offers, with 55.8% "yes" vs. 44.2% "no". This result indicates that the introduction of the second offer could lead to a higher valuation than that which would be obtained from the first price offers alone, and that the value thus obtained would be on average between the first and second price offers.

Table 2 below shows the comparison of "yes" and "no" responses to the two price offers in relation to the characteristics of the respondents. The acceptance rate for both offers (26.9% of contributors) is higher than the refusal rate (23.6% of contributors). This may explain the sensitivity of Dakar residents to improved air quality.

Individuals who accepted the offers ("yes-yes") tend to perceive air quality as bad, hence the desire to improve its quality that translates into higher valuation, unlike those who refused the two offers ("no-no"), where 1% of individuals have a good perception. They are more representative of men than of women in relation to the refusal ("no-no"). More than half of those who want to contribute to policy funding and who accepted both offers are employed (58%) and the rest (42%) are among the non-participants in the labor market and unemployed. They accept the payment options with 72.2% wanting to make a voluntary contribution and the rest accepting the taxes. This shows a preference of individuals for the voluntary contribution. This observation is not consistent with the individuals who refused the two offers, where there is a preference for taxes as a means of contribution to financing. In addition, the heads of households are representative of both those who accepted the two offers ("yes-yes") and those who refused ("no-no").



**Table 2.** Differences in responses to offers according to individual characteristics.

| Characteristics | Modalities | « Yes-Yes » | | « No-No » | |
|---|---|---|---|---|---|
| Price Offers (All three bids (CFA Francs 2000, 3500 and 5000) are combined.) | Price 1 | 81 | 26.9% | 71 | 23.6% |
| | Price 2 | | | | |
| Payment modes | Tax/electricity | 6 | 7.4% | 9 | 12.7% |
| | Tax/consumption | 7 | 8.6% | 22 | 31.0% |
| | Tax/fuel | 9 | 11.1% | 27 | 38.0% |
| | Voluntary contribution | 59 | 72.8% | 13 | 18.3% |
| Expected gains in life extension | 3 months | 32 | 39.5% | 10 | 14.1% |
| | 6 months | 20 | 24.7% | 23 | 32.4% |
| | 9 months | 29 | 35.8% | 38 | 53.5% |
| Gender | Man | 53 | 65.4% | 45 | 63.4% |
| Household head | Yes | 56 | 69.1% | 54 | 76.1% |
| Air quality related illness | Yes | 23 | 28.4% | 41 | 57.7% |
| Negative perception of air quality | Yes | 81 | 100% | 68 | 95.8% |
| Seasonality (2019) | February | 13 | 16.0% | 7 | 9.9% |
| | March | 18 | 22.2% | 21 | 29.6% |
| | April | 22 | 27.2% | 27 | 38.0% |
| | May | 28 | 34.6% | 16 | 22.5% |
| Occupation status | Occupied (worker) | 47 | 58.0% | 44 | 62.0% |
| | Inactive | 16 | 19.8% | 21 | 29.6% |
| | Unemployed | 18 | 22.2% | 6 | 8.5% |
| Departments | Dakar | 21 | 25.9% | 32 | 45.1% |
| | Guédiawaye | 8 | 9.9% | 6 | 8.5% |
| | Pikine | 48 | 59.3% | 18 | 25.4% |
| | Rufisque | 4 | 4.9% | 15 | 21.1% |

Source: Authors' calculations based on data from a survey they conducted in 2019.

## 4. Estimation Results and Discussions

### 4.1. Willingness to Pay for Improved Air Quality

Table 3 provides the results of the willingness-to-pay (WTP) estimates and shows how it varies with potential gains in life expectancy, payment options, and seasonality.

On average, consumers place a high value on improved air quality, with a willingness to pay CFA Francs 3114.8 (USD 5.7) per month per adult (or a yearly CFA Francs 37,366.6; USD 68.4). This estimate takes into account control variables, which are mostly related to individual profiles. By ignoring these individual characteristics, the average WTP becomes twice as high. Taking these controls into account is likely to increase the precision of the estimates, which in this case corresponds to a correction of an overestimation bias in WTP.

The median WTP value of CFA Francs 1532.7, far below the average, suggests a left-skewed distribution of individual valuations. This is indicative of a relatively large number of individuals willing to pay relatively little (relative to the average) for improved air quality, or of a relatively small faction of the population valuing air quality at high levels.

These estimates are generally in the same order of magnitude as the average results in the literature, particularly in the context of the African urban environment. For example, in the case of Nairobi, Kenya, ref. [16] estimated an average WTP that ranged from USD 1.88 to 3.89, following different framing methods. For the city of Cotonou (Benin), ref. [17] found an average WTP associated with the psychological cost of mortality of CFA Francs 5924 (USD 10.8) per month per adult.

**Table 3.** Willingness-to-pay estimates (CFA Francs/month).

| | | Point Estimate | Confidence Interval (95%) | |
|---|---|---|---|---|
| Average WTP | With control variables | 3114.79 *** | 1862.3 | 4367.3 |
| | Without control variables | 6387.75 *** | 5907.3 | 6868.1 |
| Median WTP | | 1532.74 ** | 317.6 | 2747.92 |
| Average WTP by payment method | | | | |
| Tax on the electricity bill | | 3581.35 *** | 2218.6 | 4944.1 |
| Tax on consumer goods | | 3721.39 *** | 2344.0 | 5098.8 |
| Fuel tax | | 3149.48 *** | 1809.8 | 4489.2 |
| Voluntary contribution | | 3608.99 *** | 2242.7 | 4975.3 |
| Average WTP according to seasonality | | | | |
| February | | 2881.19 *** | 1753.1 | 4009.3 |
| March | | 4554.66 *** | 3050.8 | 6058.5 |
| April | | 2465.49 *** | 1171.9 | 3759.1 |
| May | | 3719.88 *** | 2743.5 | 4696.3 |
| Average WTP according to life expectancy gain | | | | |
| Life expectancy gain 3 months | | 4124.79 *** | 2634.4 | 5615.2 |
| Life expectancy gain 6 months | | 2959.15 *** | 1600.3 | 4318.0 |
| Life expectancy gain 9 months | | 4011.40 *** | 2518.5 | 5504.3 |

Notes: Control variables are the characteristics presented in the appendices. The significance level is indicated by *** (1%) and ** (5%).

In terms of payment options, the results show that the opportunity to contribute through a tax on consumer goods is associated with a higher willingness to pay. Taken together, voluntary contribution opportunities correspond to a WTP that is in the high brackets, compared to coercive contributions (through a tax of some sort). This result appears to be consistent with the literature that indicates that individuals who face a voluntary payment tend to offer higher contributions, which may undermine the credibility of self-reported values [18]. However, the fact that WTPs across options are not significantly different from each other, given the overlapping confidence intervals, limits the extent of such bias.

For the differentiation according to seasonality, individuals are willing to pay more during the month of March, which corresponds to the beginning of the hot period, which is also accompanied by dust in most of the territory.

The willingness to pay for a gain in life expectancy is higher when the expected gain is 3 months than if it is 6 or 9 months. This result may suggest that individuals consider the expected gain in life expectancy uncertain and not very credible beyond 3 months.

Table 4 provides information on the main results of the value of life year (VOLY) calculations, based on WTP for the 3, 6, and 9 month gains. For an improvement in air quality, the average gain is CFA Francs 44,502.8 (USD 80.9) per adult. This monetary value, corresponding to an additional year of life associated with improved living conditions (i.e., less pollution), is higher for the 3-month life expectancy gain offer than for those corresponding to 6 or 9 months, although the differences are not statistically significant. However, these values appear to be very low, compared, for example, to CFA Francs 142.8 million (USD 0.26 million) for the city of Cotonou [17], or those by [19] of GBP 11,826 (USD 8760) for a 3-month expected gain and GBP 17,869 (USD 13,251) for 6 months. However, according to the latter authors, the validity approach could help to compare the results of existing studies in the literature, but any differences or similarities are difficult to interpret unambiguously, given the confounding results from different methods and the person-specific uncertainty.

**Table 4.** Monetary value of life expectancy gains (CFA Francs).

| Life Expectancy Gains | Value of Life Year (VOLY) | | |
|---|---|---|---|
| | Point Estimate | 95% Confidence Interval | |
| 3 months | 49,578.2 *** | 31,693.4 | 67,463.1 |
| 6 months | 35,550.2 *** | 19,243.5 | 51,856.9 |
| 9 months | 48,163.8 *** | 30,249.4 | 66,078.1 |

Note: The stars *** denote significance at 1%.

## 4.2. Determinants of Willingness to Pay

Table 5 presents the results obtained from the estimation of the bivariate probit model. The level of the first price offer is decisive for the final valuation of air quality, and its effect is negative: the higher it gets, the lower the WTP. This is consistent with the standard law of demand, and is found throughout the literature (see for instance [20]).

**Table 5.** Estimation results of the determinants of willingness to pay.

| Variables | "Yes" Response to the First Bid | | "Yes" Responses to the Second Bid | | dy/dx(.) |
|---|---|---|---|---|---|
| | Coefficients | Std Errors | Coefficients | Std Errors | |
| First price offer | −0.001 *** | 0.001 | | | 0.001 *** |
| Age | 0.007 | 0.010 | 0.001 | 0.009 | 0.002 |
| Education | 0.050 *** | 0.019 | 0.004 | 0.016 | 0.012 *** |
| Household size | −0.036 * | 0.021 | −0.022 | 0.021 | −0.012 * |
| Air quality perception | 2700 *** | 0.664 | 0.46 | 0.577 | 0.681 *** |
| Gender (reference: male) | −0.151 | 0.189 | −0.095 | 0.193 | −0.051 |
| Household head | −0.332 | 0.259 | −0.115 | 0.246 | −0.095 |
| Air quality related illness | 0.078 | 0.203 | −0.161 | 0.185 | −0.013 |
| Sick family member | −0.352 | 0.252 | 0.074 | 0.224 | −0.063 |
| Proportion of children under 5 years old | −0.005 | 0.007 | −0.003 | 0.007 | −0.002 |
| Income | 0.211 * | 0.108 | 0.053 | 0.091 | 0.056 * |
| Life expectancy gain (reference: 9 months) | | | | | |
| 3 months gain | 0.281 | 0.291 | 0.410 ** | 0.189 | 0.140 ** |
| 6 months gain | −0.031 | 0.235 | 0.285 | 0.19 | 0.047 |
| Departement (reference: Dakar) | | | | | |
| Guédiawaye | −0.110 | 0.315 | −0.062 | 0.281 | −0.036 |
| Pikine | 0.358 | 0.234 | −0.043 | 0.213 | 0.071 |
| Rufisque | 0.168 | 0.327 | −0.681 ** | 0.299 | −0.093 ** |
| Months (reference: February) | | | | | |
| March | −0.632 ** | 0.274 | −0.168 | 0.247 | −0.171 ** |
| April | −0.859 *** | 0.290 | −0.297 | 0.247 | −0.245 *** |
| May | 0.036 | 0.260 | 0.22 | 0.244 | 0.05 |
| Payment options (reference: voluntary contribution) | | | | | |
| Tax on the electricity factor | −1393 *** | 0.323 | −0.768 *** | 0.267 | −0.453 *** |
| Tax on consumer goods | −1.086 *** | 0.279 | −0.770 *** | 0.247 | −0.386 *** |
| Fuel tax | −1.393 *** | 0.251 | −0.846 *** | 0.231 | −0.467 *** |
| Occupation status (reference: employed) | | | | | |
| Inactive | −0.243 | 0.249 | −0.091 | 0.229 | −0.071 |
| Unemployed | −0.132 | 0.296 | −0.074 | 0.289 | −0.043 |
| Constante | −0.502 | 0.967 | 0.417 | 0.771 | |

N = 301
Pseudo-R$^2$ = 0.224
Wald chi2 (47) = 166.89 ***
Log likelihood = −321 904
Independence test (Wald): chi2 = 11.865 ***

Notes: The variables explained are the dichotomous responses to the first and second price offers. Standard deviations are robust (corrected for heteroskedasticity) and significance levels at 1, 5, and 10% are represented by ***, **, and *.

The level of education also appears to be significant: the higher the level of education, the higher the valuation levels. Educated individuals are thus more sensitive to environmental issues and tend to have a better grasp of the stakes involved, particularly for health. This result, which positively links education and the valuing of the environment, is almost general in the literature (see for example [17]).

Household size is also important in individual air quality valuations. However, its effect is negative, suggesting that larger households value the environment less than smaller households. A large household would have many burdens, all else being equal, and thus

be less willing to pay for an additional burden from improved air quality, although they may well place a significant value on the good.

Individuals' perceptions of air quality play a role in the value they attach to its attributes. Indeed, individuals pay more if they perceive the air they breathe to be of poor quality. Individuals feel much more exposed to pollution, hence the desire to pay to improve its quality. This result is consistent with much of the literature (see [19,21], for instance).

Household income is a significant contributing factor with a positive effect. This is in line with economic theory, which states that income and quantity consumed are positively associated with the demand for goods. In other words, high-income individuals tend to diversify their consumption basket, and in this diversification strategy, they tend to choose products of improved quality. In this sense, they will better value and consume improved quality air. This result is similar to those of [19] and [17].

Perceptions in terms of life expectancy gains due to healthier air quality contribute to the valuation of the good by individuals. Only the minimum expected gain of 3 months is significant, however. Ref. [22] attained this result in estimating a monetary value for life expectancy gains from pollution control under specific conditions regarding the information set. However, there is no significant difference for valuation levels between the 9-month and 6-month life expectancy gain. The higher life expectancy gains (6 or even 9 months) are much more associated with uncertainty, which limits their credibility.

There is also spatial differentiation in air quality valuations. The department of Rufisque is associated with a lower valuation than the other departments in the region. This result could be explained by the fact that individuals living in this department developed certain resilience mechanisms that allow them to live with pollution in such a way that they value an improvement in its quality less. Between the departments of Dakar, Pikine, and Guédiawaye, there is no significant difference.

The results also indicate a seasonal variation in valuations, with the months of March and April corresponding to relatively lower valuation levels than February and May. While February is often marked by episodes of dust storms, May corresponds most often to the peak of the heat that heralds the rainy season starting in June. Exposed to climatic conditions when they are most unfavorable, individuals thus tend to better appreciate their attributes, which they can then easily translate into monetary value.

Valuation levels are also determined by the payment vehicles considered. The prospect of voluntary contribution is associated with greater willingness to pay compared to taxes in various forms. This result is almost universal in the literature. It is generally accepted that individuals tend to overestimate their willingness to pay when there is no coercive commitment, which limits the scope of willingness to pay in reality and raises the problem of credibility of responses [18]. It goes without saying, then, that from the perspective of public contributions to the financing of air quality improvement policies, various payment modalities could be envisaged.

## 5. Conclusions

The paper was concerned with analyzing individual attitudes towards air pollution and the extent to which air quality is valued in the context of urban Africa, particularly in Dakar, Senegal. The results clearly indicate that, overall, despite a relatively high level of concentration of hazardous pollutants, such as surface ozone, dust particles, carbon monoxide, sulphur dioxide, and nitrogen dioxide, concern about its adverse effects is real. In fact, African populations do tend to be no more and no less averse to air pollution and its adverse economic and health impacts than their counterparts elsewhere in the world. This shows not only in their relatively high perception of air quality, but also in their willingness to pay for its improvement. This is so despite a great deal of heterogeneity in individual valuation schemes that form the basis for targeted policy initiatives to not only further raise public awareness of air pollution (by getting individuals unwilling to contribute onboard), but also to have greater implication into both funding and implementation

of environmental measures (by focusing on specific payment vehicles and the timing or seasonality, for instance).

These results, by offering an understanding of the economic processes of air pollution from the demand side, are indeed indicative of existing support among the general populations for public policies aiming at combatting air pollution, in addition to revealing the extent of their legitimacy. Such a consciousness about air quality degradation, on one hand, and willingness to financially contribute to targeted policy initiatives geared towards improving air quality, on the other hand, are welcome inputs to the efforts to better reconcile economic, environmental, and social processes in order to achieve sustainable development. To that end, future research accounting for the differentiated spatial and temporal incidence of air pollution and the severity of its adverse effects would bring additional insights into the collective goal of making the quest for economic development environmentally friendly.

**Author Contributions:** Data curation, S.M.D.; Formal analysis, A.S.; Investigation, S.M.D.; Methodology, A.S.; Supervision, A.S.; Writing—original draft, S.M.D. All authors have read and agreed to the published version of the manuscript.

**Funding:** This research was funded by the African Economic Research Consortium (AERC) as part of the Collaborative Research Program on "Climate Change and Economic Development in Africa" (grant number RC21583).

**Institutional Review Board Statement:** Ethical review and approval were waived for this study due to the nature of the survey that was anonymous and the collected information that was not sensitive.

**Informed Consent Statement:** Informed consent was obtained from all interviewees involved in the study, and who had been told that the purpose of the survey was a scientific research.

**Data Availability Statement:** Data collected by means of survey, as well as the Stata codes used to generate the results, are available upon request.

**Acknowledgments:** The authors are thankful to the AERC, and to Mamadou Lamine Paye, a statistician, who assisted with the survey in Dakar, Senegal. All individuals included in this section have consented to the acknowledgement.

**Conflicts of Interest:** The authors declare no conflict of interest. The funders had no role in the design of the study; in the collection, analyses, or interpretation of data; in the writing of the manuscript; or in the decision to publish the results.

## Appendix A

**Table A1.** Detailed descriptive statistics of the variables.

| Variables | Obs. | Average/Frequency | Min. | Max. |
|---|---|---|---|---|
| Perception of air quality 1 = polluted, 0 = not polluted | 427 | 0.91 | 0 | 1 |
| Air quality rating 1 = "very poor" to 10 = "excellent" | 427 | 5.01 | 1 | 10 |
| Willingness to pay/contribute 1 = yes | 427 | 0.70 | 0 | 1 |
| For/against the principle of payment 1 = for, 0 = against | 126 | 0.29 | 0 | 1 |
| Gender: 1 = male | 427 | 0.68 | 0 | 1 |
| Age (years) | 427 | 47.22 | 18 | 80 |
| Education (years) | 427 | 5.88 | 0 | 18 |
| Monthly income (CFA Francs 1000): 1 = 0–99, 2 = 100–199, 3 = 200–299, 4 = 300–399, 5 = 400–499, 6 = 500 | 427 | 1.99 | 1 | 6 |
| Head of household status: 1 = yes | 427 | 0.72 | 0 | 1 |
| Household size (count) | 427 | 9.28 | 1 | 25 |

**Table A1.** *Cont.*

| Variables | Obs. | Average/Frequency | Min. | Max. |
|---|---|---|---|---|
| Number of children (<=5 years, count) | 427 | 1.63 | 0 | 8 |
| Family member suffered from air pollution-related illness: 1 = yes | 427 | 0.47 | 0 | 1 |
| Payment vehicle: tax on electricity bill: 1 = yes | 427 | 0.1 | 0 | 1 |
| Payment vehicle: tax on consumer goods: 1 = yes | 427 | 0.2 | 0 | 1 |
| Payment vehicle: fuel tax (%) 1 = yes | 427 | 0.2 | 0 | 1 |
| Payment vehicle: voluntary contribution (%) 1 = yes | 427 | 0.2 | 0 | 1 |
| Labor market status: employed: 1 = yes | 427 | 0.6 | 0 | 1 |
| Labor market status: inactive: 1 = yes | 427 | 0.3 | 0 | 1 |
| Labor market status: unemployed: 1 = yes | 427 | 0.1 | 0 | 1 |

Source: Authors' calculations based on data from a survey they conducted in 2019.

## Appendix B

**Table A2.** Distribution of (strictly) positive willingness to pay or to contribute to policy funding.

| Responses | Yes | No | Differences |
|---|---|---|---|
| Shares | 0.70 | 0.30 | 0.40 *** |
| Age (years) | 45.70 | 50.83 | −5.13 *** |
| Gender: male = 1 | 0.66 | 0.71 | 0.05 |
| Education (# years) | 6.38 | 4.71 | 0.60 *** |
| Perception of air quality | 0.98 | 0.75 | 0.22 *** |
| Pollution-related illness | 0.52 | 0.35 | 0.18 *** |
| Income | 2.10 | 1.73 | 0.37 *** |
| HH size (count) | 8.85 | 10.29 | −1.44 *** |
| Occupied | 0.68 | 0.50 | 0.18 *** |
| Dakar | 0.43 | 0.25 | 0.17 *** |
| Guediawaye | 0.09 | 0.10 | 0.00 |
| Pikine | 0.36 | 0.40 | −0.05 |
| Rufisque | 0.12 | 0.25 | −0.12 *** |
| February | 0.20 | 0.35 | −0.15 *** |
| March | 0.25 | 0.25 | 0.00 |
| April | 0.25 | 0.25 | 0.01 |
| May | 0.15 | 0.30 | −0.15 *** |

Note: The stars *** denote the 1-percent significance level of the *t*-test difference. Source: Authors' calculations based on data from a survey they conducted in 2019.

## Appendix C

**Table A3.** Estimation results of the bi-variate probit model (second stage).

| Variables | Single Bound | | Double Bound | |
|---|---|---|---|---|
| | Coeff. | Stand. Err. | Coeff. | Stand. Err. |
| First price offer | 0.001 *** | 0.000 | | |
| Age | −0.014 | 0.020 | −0.007 | 0.017 |
| Years of education | 0.068 *** | 0.024 | 0.011 | 0.020 |
| Household size | −0.049 ** | 0.023 | −0.027 | 0.023 |
| Perception of air quality | 4942 *** | 1.693 | 1.312 | 1.468 |
| Gender (reference: male) | −0.534 | 0.356 | −0.245 | 0.338 |
| Head of Household | −0.137 | 0.303 | −0.038 | 0.278 |
| Air-pollution-related illness (self) | 0.388 | 0.323 | −0.036 | 0.293 |
| Air-pollution-related illness (family member) | 0.000 | 0.377 | 0.211 | 0.318 |
| Proportion of children under 5 years old | −0.012 | 0.009 | −0.006 | 0.008 |
| Income | 0.469 ** | 0.231 | 0.155 | 0.198 |
| Life expectancy gain (Reference: 9 months) | | | | |
| 3-month gain | 0.443 | 0.324 | 0.472 *** | 0.217 |

**Table A3.** *Cont.*

| Variables | Single Bound | | Double Bound | |
| --- | --- | --- | --- | --- |
| | **Coeff.** | **Stand. Err.** | **Coeff.** | **Stand. Err.** |
| 6-month gain | −0.008 | 0.235 | 0.286 | 0.191 |
| Department (Reference: Dakar) | | | | |
| Guédiawaye | −0.168 | 0.316 | −0.086 | 0.282 |
| Pikine | 0.434 * | 0.253 | −0.010 | 0.226 |
| Rufisque | 0.143 | 0.324 | −0.689 ** | 0.300 |
| Month (Reference: February) | | | | |
| March | −0.002 | 0.551 | 0.077 | 0.494 |
| April | −0.449 | 0.416 | −0.140 | 0.372 |
| May | 0.641 | 0.493 | 0.453 | 0.463 |
| Payment Method (Reference: Voluntary Contribution) | | | | |
| Tax on the electricity factor | −1382 *** | 0.326 | −0.763 *** | 0.268 |
| Tax on consumer goods | −1.060 *** | 0.279 | −0.764 *** | 0.247 |
| Tax on fuel | −1.376 *** | 0.254 | −0.838 *** | 0.233 |
| Labor market status (Reference: Employed) | | | | |
| Non-participant | −0.531 | 0.335 | −0.199 | 0.313 |
| Unemployed | −0.046 | 0.300 | −0.042 | 0.292 |
| Inverse Mill's ratio | 2406 | 1.763 | 0.931 | 1.634 |
| Constant | −3862 | 2593 | −0.857 | 2236 |

N = 301
Pseudo-$R^2$ = 0.2267
Wald chi2 (49) = 170.74 ***
Log likelihood = −320,772
Wald test of rho = 0: chi2 (1) = 11.818 ***

Notes: Explained variables are dichotomous responses to first and second price offers. Standard errors are robust (corrected for heteroskedasticity) and significance levels at 1, 5, and 10% are represented by ***, ** and *.

## Appendix D

**Table A4.** Determinants of willingness to pay—first step of the Heckman procedure.

| Variables | Coeff. | Stand. Err. |
| --- | --- | --- |
| Age | −0.022 *** | 0.008 |
| Educations (years of study) | 0.018 | 0.015 |
| Perception of air quality | 1.415 *** | 0.295 |
| Gender (reference: male) | −0.400 ** | 0.180 |
| Head of household | 0.193 | 0.220 |
| Air-pollution-related illness (self) | 0.322 * | 0.176 |
| Air-pollution-related illness (family member) | 0.332 | 0.203 |
| Income | 0.279 *** | 0.092 |
| Proportion of children under 5 years old | −0.007 | 0.006 |
| Life expectancy gain (Reference: 9 months) | | |
| 3-month gain | 0.144 | 0.176 |
| 6-month gain | 0.003 | 0.177 |
| Department (Reference: Dakar) | | |
| Guédiawaye | −0.070 | 0.244 |
| Pikine | 0.072 | 0.196 |
| Rufisque | −0.057 | 0.243 |
| Month (Reference: February) | | |
| March | 0.614 *** | 0.216 |
| April | 0.355 * | 0.199 |
| May | 0.583 *** | 0.201 |
| Reference: Employment Status (Employed) | | |
| Non-participant | −0.254 | 0.195 |
| Unemployed | 0.060 | 0.247 |
| Household size | −0.012 | 0.020 |
| Constant | −0.715 | 0.523 |

N = 427
LR Chi2 (20) = 89.34 ***
Pseudo $R^2$ = 0.2166
Log likelihood = −202.9321

Notes: results of probit model estimation (standard errors are robust). The significance levels at 1, 5, and 10% are represented by ***, ** and *.

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
