# Peer review of "Air Pollution in Urban Africa: Understanding Attitudes and Economic Valuation in the Case of Dakar, Senegal"

_sustainability, doi:10.3390/su15021494_

Round 1

Reviewer 1 Report

The article is well prepared and written. 

Author Response

The comments from Reviewer 1 are very welcome. Regarding the "English language and style", it has been significantly improved in the revised version of the draft paper.

Reviewer 2 Report

This manuscript describes the attitudes and willingness to pay for air quality improvement for people in Dakar, Sinegal. Overall the manuscript was OK, however, there are a few issues and comments for improvement so that the manuscript will be good for publication. The comments/ issues are listed below:

1. Title - Basically, the title did not describe the manuscript. The part of air quality in the context of climate change was not discussed in this manuscript. The authors only discusses the attitude and willingness to pay for better air quality.

2. Introduction - The objective written here was not the main findings in this manuscript. It was said that the objective was "to analyze the benefits associated with improving air quality in Dakar, in a context of global warming" (line 68-69) however, no results on this objective was given and discussed. In addition, the author should highlight what are the effects of climate change experienced in Senegal in this recent years because Figure 1 only gives the air quality index in Dakar and this does not show any sign of increase air pollutant in Senegal. Is Senegal really experiencing climate change/ global warming such as typhoon, increase in temperature, extreme raining etc?

3. Study context - The authors should include how many monitoring stations were included or presented in this study. What are the range of air quality index used in Senegal? As I highlighted above, does Senegal/ Africa is facing any of the sign of global warming? Some of the references were not listed in the reference list. Some of them were more than 10 years. Recent references should be included.

4. Methods - no references for the probability distribution/ function in the method. The purpose of conducting the analysis should be included.

5. Results- Firstly, the authors should include the distribution percentage of the person that answered the questionnaires. For example the gender, the income etc. Other than that, the relation between the methods given in the method and results were not clear. In addition, no to very minimal discussion given on air quality to the context of global warming as mentioned objective in the introduction.

6. Conclusion - The authors tend to write the conclusion according to personal perception rather than supported by the results. The results of air quality presented in this study was just an air quality index in Dakar for almost 9 years period. Hence, in my observation, the authors misunderstood the concept of global warming and only discusses air pollution instead.

The details of the comments is given in the attached document.

Reviewer 3 Report

The authors use questionnaire data to explored the willingness to pay for air quality under the background of climate change. The research is valid and innovative. However, there are have some problems need to be solved as follow:

Major problems 

1.There are four departments in Dakar(line332).What's the difference of residents' willingness to pay for air quality between four depatments?

2.The authors should give some suggestions to improve the willingness to pay for air quality  and public participation. How to coordinate the different willingness to pay for air quality.

3.The conclusions should show the core opinion in this manuscript. The authors should conclude several points according to estimation results. The perspective of core opinion mainly included household size, seasonality , level of education, high income  and so on. At present, the sentences in  conclusion mainly include background and significant, it is lack of core opinion.

4.The structure of this manuscript is a little mess. I suggest integrate section 2 study context and  section 3 methodology and survey data into one section. And section 4 Estimation results and discussions should divide into two sections, one section is estimation results,another section is discussion. The discussion should include the study innovation and significance, countermeasures and suggestions,outlook three sub-sections.

Minor problems:

1.In Introduction, the authors illustrated the contribution of climate change to human health(line 24-25). I think the relationship of climate and people should also be introduced. Authors should  refer to the paper as follow. Spatiotemporal relationship characteristic of climate comfort of urban human settlement environment and population density in China.Front.Ecol.Evol.2022,10,953725.doi:10.3389/fevo.2022.953725.

2.The repeated sentences should be revised or deleted. For instance, the sentence in line 81-89 is similar to abstract(line 14-19).

3.The presentation is not intact enough. For example, the survey was conducted over a four-month period (February-May 2019) to capture the seasonal variability of the pollution phenomenon(line 79-80).According the line 95-96, the survey was conducted to capture the dry seasonal variability of the pollution phenomenon. Futhermore, the authors illustrated Dakar produces 2500 tons of garbage per day(CSE,2013),and not all of it is taken to the disposal sites. The authors should present where are the rest of garbage. The degrade air quality beyond the thresholds commonly accepted by the WHO in most cases, and even those set at national level(line113-114).  The authors should illustrated the specific thresholds of WHO and national level.

4.In line 125,I understand that four subdivisions refer to Bon, Moven,Mauvais and Tres Mauvais. If so, please illustrate it clearly.

5.According to OMS, the Ministry of Health and Social Action had estimated 3500 cases of patients suspected of having asthma and over 7000 cases in 2014(line 147-148). It refer to Africa or Senegal. 

6.The data source should be illustrated more detailed. The actual survey was conducted monthly between February and May 2019. In total, a sample of 427 people was obtained. It is representative of the population of Dakar through its four departments. How many samples are obtained in each of departments and month? In addition, how many data are obtained before clean, and how many data are remained after clean? It should be illustrated detailed.

7. The authors should compare the residents' willingness to pay for air quality who are live in different land use area in furture study. For example, the residents live in the industrial area maybe have high willingness to pay for air quality, while the residents who live near the nature reserves maybe have low willingness to pay for air quality. Because the different urban land area could  affect the PM 2.5. The authors should refer to the article as follow. Spatial and temporal heterogeneity of urban land area and PM2.5 concentration in China.Urban Climate,2022,45,101268.doi:https://doi.org/10.1016/j.uclim.2022.101268.

8.The English need to be polished.

Round 2

Reviewer 2 Report

Generally the manuscript was better than the first version esppecially on the direction of overall manuscript. I have a few issues regarding this edited version:

1. The first paragraph of introduction still describing about climate change while, the whole manuscript is only explaning about air pollution. I think this part should also be revised and just generally explain on air pollution worldwide.

2. The equations of distribution in the method still no reference.

3. The sample size for the survey was away too little if compared to the total population (143 samples compared to 3.52 million?)

Author Response

Response to Reviewer 2 Comments (round 2)

Generally the manuscript was better than the first version esppecially on the direction of overall manuscript. I have a few issues regarding this edited version:

  1. The first paragraph of introduction still describing about climate change while, the whole manuscript is only explaning about air pollution. I think this part should also be revised and just generally explain on air pollution worldwide.

Response: I agree, the first paragraph has been thoroughly rewritten, and it now extends to two paragraphs. The discussion now focuses on pollution as an environmental issue and a public health crisis, both around the world and in Africa.

  1. The equations of distribution in the method still no reference.

Response: The reference of Hanemann (1984) is provided (see line after equation 3); the author has suggested various development of the initial work of McFadden (1974) that laid out the theory behind the random utility model.

  1. The sample size for the survey was away too little if compared to the total population (143 samples compared to 3.52 million?)

Response: The number 143 in Table 1 refers to the number of interviewees who responded “yes” to the first bid. The whole sample include these individuals, plus those who answers “no” (158), and those who are not willing to contribute (126). In total, the actual sample size is 427 individuals, which is obtained through the standard formula that accounts for the population of the region and its proportion, the margin of error, and the Z score.

Reviewer 3 Report

The quality of the manuscript has been improved significantly. However, there still have several minor problems need to be revised.

1. What is the meaning of ss of 2021 in line 144.

2.The authors introduced the level of pollutants detailed in line 160-168. However, it is not related to this manuscript. I suggest delete this redundant section.

3. The 10 should be superscripted in line 215.

4. In discussing the scope of finding in the conclusion, the authors mentioned the possibility of further research that would delve into additional sources of heterogeneity in individual attitudes and valuation. However, it is lack of existing researches as foundation. The relevant reference should be cited as a foundation as follow. Spatial and temporal heterogeneity of urban land area and PM2.5 concentration in China.Urban Climate,2022,45,101268.doi:https://doi.org/10.1016/j.uclim.2022.101268.

5. The format of references should be revised. Please refer to the template of Sustainability strictly.

Author Response

Response to Reviewer 3 Comments (round 2)

The quality of the manuscript has been improved significantly. However, there still have several minor problems need to be revised.

  1. What is the meaning of ss of 2021 in line 144.

Response: This is a typo that has been corrected

2.The authors introduced the level of pollutants detailed in line 160-168. However, it is not related to this manuscript. I suggest delete this redundant section.

Response: This has been introduced following a comment from another anonymous referee as a way to allow a better reading of the air quality index (i.e. figure 1)

  1. The 10 should be superscripted in line 215.

Response: This has been corrected

  1. In discussing the scope of finding in the conclusion, the authors mentioned the possibility of further research that would delve into additional sources of heterogeneity in individual attitudes and valuation. However, it is lack of existing researches as foundation. The relevant reference should be cited as a foundation as follow. Spatial and temporal heterogeneity of urban land area and PM2.5concentration in China.Urban Climate,2022,45,101268.doi:https://doi.org/10.1016/j.uclim.2022.101268.

Response: That part of the conclusion has been slightly rewritten (parag. 2)

  1. The format of references should be revised. Please refer to the template of Sustainability strictly.

Response: The format has been updated.